# A Third-Person Appraisal Agent: Learning to Reason About Emotions in Conversational Contexts

## Abstract

Emotion reasoning is crucial for achieving human-like emotional understanding in Emotion Recognition in Conversation (ERC). Current ERC datasets provide only emotion-labeled utterances, lacking the rich annotations necessary for emotion reasoning. Although Large Language Models (LLMs) show promise in generating rich emotional knowledge, they still struggle to apply this knowledge effectively for emotion reasoning. To address these challenges, we propose a learning framework based on cognitive appraisal theory, utilizing an agent powered by LLMs to learn emotion reasoning from a third-person perspective, which we refer to as the third-person appraisal agent. This learning framework comprises two phases: self-evaluation and meta-evaluation. In the self-evaluation phase, the agent generates appraisals essential for inferring emotions, incorporating counterfactual thinking to refine its appraisals. The meta-evaluation phase uses reflective actor-critic reinforcement learning to train the agent to generate accurate appraisals during testing. The training samples are appraisals generated during the self-evaluation phase, which eliminates the need for human annotations. By fine-tuning a specialized LLM in this framework, we significantly outperform baseline LLMs across ERC tasks, demonstrating superior reasoning capabilities and better generalization across various dialogue datasets. Additionally, we provide interpretable results that clarify the model's reasoning process behind its predictions. To the best of our knowledge, this research is the first to apply cognition-based methods to enhance LLMs' emotional reasoning capabilities, marking a significant advancement toward achieving human-like emotional understanding in artificial intelligence. The code is available here.

## 1 Introduction

Emotion reasoning is crucial for understanding the causes behind expressed emotions, as it involves analyzing the complex interplay of a speaker's thoughts, feelings, and behaviors in the field of Emotion Recognition in Conversation (ERC)(Wondra & Ellsworth, 2015; Ong et al., 2019). Applications of ERC range from mental health support systems to empathetic conversational systems, where emotion reasoning is essential for advancing toward human-like conversations. Current ERC methods (Wondra & Ellsworth, 2015; Ribeiro et al., 2016; Hazarika et al., 2018; Ong et al., 2019; Jiao et al., 2020; Vellido, 2020; Gao et al., 2021; Hu et al., 2021a; Li et al., 2022; Sabour et al., 2022; Zhao et al., 2022; Cortiñas-Lorenzo & Lacey, 2023; Hu et al., 2023) rely on identifying emotion triggers to infer emotions. However, these triggers are surface-level stimuli that evoke emotional reactions and fail to capture the deeper underlying reasons that explain why certain triggers lead to specific emotional responses (Poria et al., 2019; Yang et al., 2024). This gap raises a critical research question: How can we develop emotion reasoning approaches that more closely mimic human understanding of emotions in conversations?

Currently, there are no datasets specifically designed for emotion reasoning tasks (Poria et al., 2019; Gan et al., 2024). Existing ERC datasets, such as IEMOCAP (Busso et al., 2008) and DailyDialog (Li et al., 2017), only provide emotion labels for individual utterances and lack the detailed annotations needed for emotion reasoning. Large Language Models (LLMs) like GPT-3 and GPT-4 have shown potential in generating rich emotional content (Li et al., 2023; Qian et al., 2023; Team, 2024).

For instance, Zhang et al. (2023) utilizes LLMs to generate visual information, providing supplementary knowledge for emotional context, while Lee et al. (2022) leverages GPT-3's in-context learning to generate empathetic responses. However, despite their ability to produce rich emotional knowledge, LLMs still struggle to apply this knowledge effectively for emotion reasoning. Our work addresses this gap by using LLMs to generate the detailed annotations necessary for emotion reasoning in ERC tasks.

The appraisal theory of emotion (Lagattuta et al., 1997; Wondra & Ellsworth, 2015; Ong et al., 2019) explains that emotions emerge from individuals' appraisals (i.e., cognitive evaluations) of situations, particularly in relation to their goals, desires, intentions, and expectations. Inspired by this theory, we develop a novel framework that integrates cognitive appraisal principles into emotion reasoning tasks. At the core of this framework is the third-person appraisal agent, powered by LLMs. This agent acts as an external observer, analyzing conversations to evaluate how contextual utterances align with an interlocutor's objectives and expectations, and subsequently inferring their emotional reactions. For instance, as shown in Figure 1, Person A's anger may result from Person B's indifferent attitude, which contradicts Person A's expectations. By simulating the cognitive appraisal process, this approach offers a possible solution for emotion reasoning, enabling LLMs to better capture the emotional dynamics of conversational contexts.

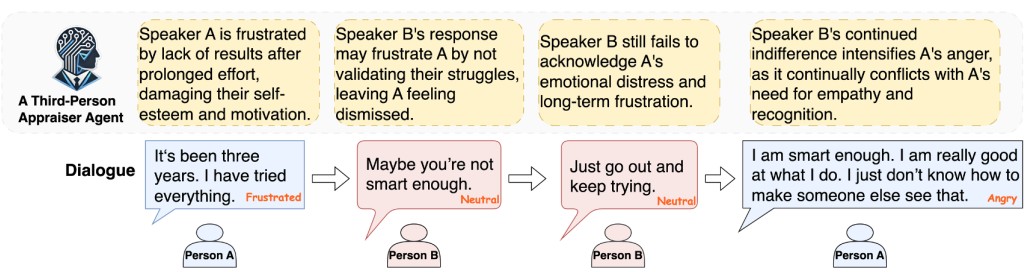

Figure 1: An example of how a third-person appraisal agent works. The example sample is drawn from the IEMOCAP dataset (Busso et al., 2008).

We introduce a novel learning framework comprising two distinct phases: self-evaluation and meta-evaluation, both enhanced by integrated reflection mechanisms. In the self-evaluation phase, the agent engages in reflective assessment by generating and refining emotional appraisals through counterfactual reasoning (Roese, 1997). This process enables the agent to explore alternative emotional responses and evaluate their alignment with conversational context. To the best of our knowledge, this work is among the first to incorporate counterfactual thinking into a verbal reinforcement learning (RL)-based method (Shinn et al., 2024) for emotion reasoning tasks. Building on this foundation, the meta-evaluation phase employs a reflective actor-critic RL strategy (Flavell et al., 2001; Haarnoja et al., 2018) to continuously refine the model's reasoning strategies based on self-generated correct and incorrect appraisals from the self-evaluation phase. This meta-evaluation phase iteratively enhances the agent's emotional understanding and reasoning accuracy without requiring human annotations.

Meanwhile, the efficient and reproducible evaluation of emotion reasoning remains challenging due to the reliance on manual annotations(Kazienko et al., 2023; Madaan et al., 2024; Huang et al., 2024), which are time-consuming, costly, and highly variable. This variability limits large-scale model comparisons and hinders the reliable replication of results. We aim to simplify emotion reasoning performance evaluation by enabling LLMs to automatically assess and score emotional reasoning tasks. Specifically, we evaluate: (1) Emotional Comprehension, which assesses the ability to recognize emotional causes and understand the speaker's motivations;(2) Contextual Understanding, which measures the understanding of context and how emotions evolve within a conversation; and (3) Expressive Coherence and Performance, which evaluates whether the model communicates its emotional reasoning clearly and is easy to understand. In this way, we transform complex emotion reasoning evaluation into a multiple-choice format that capable LLMs can assess, enabling an efficient and reproducible method for emotion reasoning evaluation in the ERC field.

Our experiments demonstrate the effectiveness of our approach, surpassing LLM baselines in both accuracy and weighted F1 scores for ERC tasks. To further validate its generalization capabilities,

We tested our model on unseen conversational contexts, where it exhibited robust and consistent performance across various scenarios, including reasoning about previously unseen emotions. The main contributions of this paper are summarized as follows:

- We propose a novel framework that integrates cognitive theory into emotion reasoning tasks, enabling LLMs to autonomously refine their reasoning processes in alignment with cognitive appraisal principles. This is the first work to enhance LLMs' emotion reasoning capabilities in ERC by guiding them to evaluate emotions based on human cognitive reasoning.

- We incorporate a reflection mechanism to enhance the model's emotion evaluation in two complementary ways. First, it utilizes counterfactual thinking to generate reflections. Second, it employs the actor-critic RL strategy to improve the model's reasoning capabilities by leveraging these reflections, which serve as a limited number of demonstration examples.

- Experimental results demonstrate that our model enhances prediction performance and generalizability across new dialogue datasets. Additionally, we design an objective method for evaluating emotion reasoning performance, focusing on emotional comprehension, contextual understanding, and expressive coherence and clarity. This evaluation provides a reproducible, explainable, and efficient alternative to manual annotations.

## 2 RELATED WORK

Current approaches to emotion reasoning with LLMs emphasize prompt tuning for tasks such as emotional cause extraction (Doe & Smith, 2023; Bhaumik & Strzalkowski, 2024; Belikova & Kosenko, 2024). However, there is limited research exploring the integration of self-reflection or feedback mechanisms specifically within emotion reasoning tasks. Currently, self-reflection or feedback mechanisms have been explored in other domains, such as mathematical reasoning, code generation, and so on (Welleck et al., 2022; Yang et al., 2022; Paul et al., 2023; Madaan et al., 2024; Shinn et al., 2024). Shinn et al. (2024) introduces Reflexion, a self-reflection mechanism that enables LLMs to improve their reasoning capabilities by learning from past mistakes. However, the application of Reflexion to emotion reasoning tasks has yet to be thoroughly investigated. Although Madaan et al. (2024) demonstrates self-reflection in sentiment style transfer, which involves modifying the sentiment of a text while preserving its meaning, this task is tangentially related to ERC tasks. Our work is unique in combining reflection-mechanism with a domain-principles-driven approach based on cognitive appraisal theory. This framework allows LLMs to not only generate self-feedback and refine their outputs but also align with human-like emotion reasoning processes, simulating how humans understand emotions.

### 2.1 PROBLEM DESCRIPTION

Given a dialog consisting of a sequence of utterances $U = \{u_1, u_2, \ldots, u_I\}$, each of which is associated with a specific speaker. The number of emotional categories $o$ varies depending on the number of emotional types in different evaluation datasets. The task is to generate an appraisal $a_i$ for each utterance $u_i$, and then infer an emotion label $\hat{y}_i$ based on this appraisal.

### 2.2 TWO LEARNING PHASES FOR THIRD-PERSON APPRAISAL AGENT

We introduce a third-person appraisal agent composed of three specialized LLMs: the Appraisal Generator, the Appraisal Evaluator, and the Third-Person Appraisal LLM. The agent utilizes two learning phases, consisting of self-evaluation and meta-evaluation, which enable it to perform emotion reasoning from a third-person perspective (see Figure 2).

**Appraisal Generator LLM:** In the self-evaluation process, the appraisal generator $M_G$ evaluates all relevant factors influencing the interlocutor's emotions, generating a series of appraisal trajectories. This LLM simulates human cognitive appraisal when reasoning about emotional states.

**Appraisal Evaluator LLM:** The Evaluator $M_E$ assesses the accuracy of these appraisals and provides feedback, upon which we assign reward values. We utilize $M_E$ to provide two types of rewards:

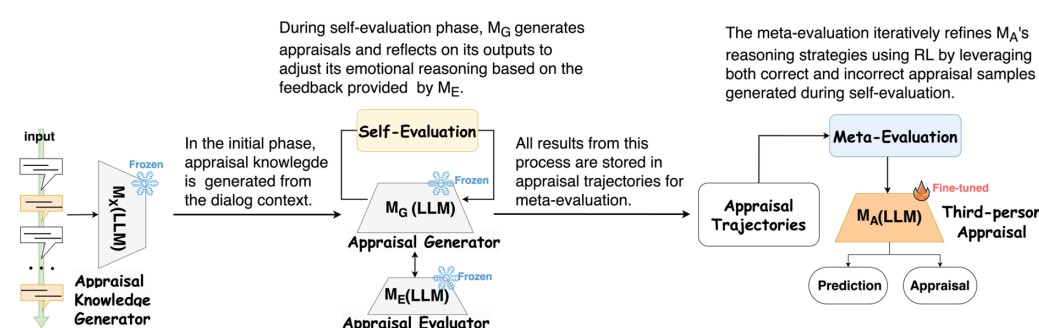

Figure 2: A workflow of the Third-Person Appraisal Agent's process

---

**Algorithm 1** VERBAL RL: SELF-EVALUATION VIA COUNTERFACTUAL REASONING

---

**Require:** Input $u_i$, dialog context $C_i$, appraisal knowledge $x_i$, models $\{M_G, M_E\}$, prompts
$\{p_a, p_c\}$, true emotion label $y_i$

1: $(a_{i,0}, \hat{y}_{i,0}) = M_G(p_a \| u_i \| C_i \| x_i)$              ▷ Initial generation (Eq.1)

2: $(r_{i,0}^{\text{actor}}, r_{i,0}^{\text{critic}}) = M_E(\hat{y}_{i,0}, y_i, a_{i,0})$              ▷ Initial feedback (Eq.2)

3: Add $(u_i, a_{i,0}, r_{i,0}^{\text{actor}}, r_{i,0}^{\text{critic}})$ to appraisal trajectory $\mathcal{D}_i$

4: **for** iteration $k = 1, 2, \ldots$ **do**

5:      $(a_{i,k}, \hat{y}_{i,k}) = M_G(p_c^k \| u_i \| x_i \| \{\hat{y}_{i,0}, \ldots, \hat{y}_{i,k-1}\})$    ▷ Counterfactual reasoning (Eq.3)

6:      $(r_{i,k}^{\text{actor}}, r_{i,k}^{\text{critic}}) = M_E(\hat{y}_{i,k}, y_i, a_{i,k})$              ▷ Feedback (Eq.4)

7:      Add $(u_i, a_{i,k}, r_{i,k}^{\text{actor}}, r_{i,k}^{\text{critic}})$ to appraisal trajectory $\mathcal{D}_i$

8:      **if** $\hat{y}_{i,k} = y_i$ **then**                  ▷ Stop condition

9:          **break**

10:     **end if**

11: **end for**

12: **return** $\mathcal{D}_i$

---

- **Action Reward** $r^{\text{actor}}$: Assigns 0 for correct emotion label predictions and -1 for incorrect ones, reinforcing accurate predictions and guiding the model to refine its appraisals.

- **Critic Reward** $r^{\text{critic}}$: Evaluates the alignment of each appraisal's valence-arousal (VA) vector with its target emotion class. Valence and arousal scores are obtained from the NRC-VAD lexicon (Mohammad, 2018) and normalized to the range [-1, 1] using min-max scaling. In the Evaluation Prompt (see Appendix B), the Evaluator $M_E$ uses the Circumplex Model (Russell, 1980) to classify emotion labels into predefined valence and arousal ranges. It then checks if the appraisals fall within these ranges, assigning a score of 0 for alignment and -1 for misalignment.

**Third-person Appraisal LLM:** During the meta-evaluation process, we fine-tune a third-person appraisal $M_A$ to optimize its appraisal strategy. This framework enables $M_A$ to generate more accurate assessments for each conversational utterance. The fine-tuned $M_A$ is then utilized to produce emotion predictions and appraisals during the inference phase. We prompt $M_A$ using an Inference-Instruction prompt (see Appendix B) to generate a predicted emotion label $\hat{y}_i$ and an appraisal $a_i$, given only the input utterance $u_i$ and the corresponding dialogue context $C_i$.

### 2.2.1 INITIAL PHASE: APPRAISAL KNOWLEDGE GENERATION

We set a window of length $l$ to gather the dialogue context $C_i$ for each utterance $u_i$. The dialogue context $C_i$ consists of the current utterance and the $l-1$ preceding utterances, each accompanied by the corresponding speaker information.

We prompt the LLM $M_X$ (e.g., GPT-4) using the AppraisalKnowledge Prompt (see Appendix B), which is designed based on the principles of cognitive appraisal theory (Watson & Spence, 2007; Ong et al., 2019), to generate appraisal knowledge $x_i$ for utterance $u_i$. This prompt enhances

appraisal-related knowledge extracted from the provided dialogue context. By leveraging true emotion labels, we generate high-quality knowledge that involves identifying key situational elements, analyzing their relevance to the speaker's goals, intentions, or expectations, and evaluating their impact on the current utterance. $u_i$. This process can be formulated as:

$$x_i = M_X(u_i, y_i, C_i)$$

The goal of generating appraisal knowledge is to enable the model to reason about emotions by evaluating how each participant's goals, desires, intentions, or expectations align with the conversational context. To achieve this, we introduce a self-evaluation phase where the model learns to generate appraisals from a third-person perspective, enhancing its ability to assess emotional dynamics through a cognitive process.

### 2.2.2 PHASE 1: SELF-EVALUATION

The self-evaluation process utilizes the appraisal generator $M_G$ to create new appraisals by adjusting its previous ones based on feedback from $M_E$ (see Algorithm 1).

The self-evaluation framework is detailed in the following steps:

We first prompt $M_G$ to generate an appraisal and an emotion label based on utterance $u_i$, dialog context $C_i$ and appraisal knowledge $x_i$. We design a AppraisalGenerator Prompt $p_a$ (see Appendix B) to achieve this:

$$(a_{i,0}, \ \hat{y}_{i,0}) = M_G(p_a \| u_i \| C_i \| x_i) \tag{1}$$

Next, we evaluate this initial appraisal and prediction with $M_E$, obtaining actor and critic rewards:

$$(r_{i,0}^{\text{actor}}, \ r_{i,0}^{\text{critic}}) = M_E(\hat{y}_{i,0}, \ y_i, \ a_{i,0}) \tag{2}$$

If the initial prediction $\hat{y}_{i,0}$ is incorrect, $M_G$ enters an iterative counterfactual reasoning loop ($k \geq 1$) to generate new appraisals. At each iteration $k$, the CounterfactualReasoning $p_c^k$ (see Appendix B) uses the history of incorrect predictions $\{\hat{y}_{i,0}, \hat{y}_{i,1}, \ldots, \hat{y}_{i,k-1}\}$ and appraisal knowledge $x_i$ to update the output for utterance $u_i$:

$$(a_{i,k}, \ \hat{y}_{i,k}) = M_G(p_c^k \| u_i \| x_i \| \{\hat{y}_{i,0}, \ldots, \hat{y}_{i,k-1}\}) \tag{3}$$

We then evaluate the updated appraisal with $M_E$:

$$(r_{i,k}^{\text{actor}}, \ r_{i,k}^{\text{critic}}) = M_E(\hat{y}_{i,k}, \ y_i, \ a_{i,k}) \tag{4}$$

This reflective process continues until the prediction is correct or a maximum number of iterations $K$ is reached. After completing the self-evaluation phase, we collect the appraisal trajectories into a replay buffer $D$:

$$D = \left\{ (u_{i,k}, \ a_{i,k}, \ r_{i,k}^{\text{actor}}, \ r_{i,k}^{\text{critic}}) \mid k = 0, \ldots, K_i; \ i = 1, \ldots, I \right\}$$

$K_i$ is the number of iterations for the $i$-th utterance. If $M_A$ makes a correct prediction at $k = 0$, we set $K_i = 0$, and the trajectory consists only of the initial appraisal.

### 2.2.3 PHASE 2: META-EVALUATION

To enhance the appraisal capability of the third-person appraisal agent, we construct a meta-evaluation process using a reflective actor-critic RL framework inspired by Zhou et al. (2024) (see Figure 3 and Algorithm 2). This framework aims to fine-tune a third-person appraisal $M_A$ via RL. In this setup, $M_A$ functions as the actor, generating appraisals for each utterance, while a critic evaluates the actor's performance and provides feedback. The iterative interaction between the actor and critic continuously refines the actor's appraisal mechanism, improving its reasoning capability.

We employ off-policy learning, the Q-function and value function are updated based on experiences sampled from a replay buffer $\mathcal{D}$, which we obtained during the self-evaluation phase. This allows the critic to learn from a broader set of experiences, improving stability and efficiency in training.

**Algorithm 2** Meta-Evaluation via Reflective Actor-Critic RL

1: **Initialize** Third-Person Appraisal $M_A$, Critics $Q_{\theta_1}$ and $Q_{\theta_2}$, Value Function $V_\psi$, and Replay Buffer $\mathcal{D}$ (an offline dataset).
2: **Initialize** Policy $\pi_\phi(a_{i,k'}|u_{i,k'})$, where $\phi = M_A$
3: Set $t \leftarrow 0$
4: **while** $t < $ T **do**
5:     Sample batch $\{u_{i,k'}, a_{i,k'}, r_{i,k'}, a_{i,k'+1}\}$ from $\mathcal{D}$.
6:     **For terminal steps** (where $k' = K_i$), **set** $a_{i,k'+1} = a_{i,k'}$.
7:     **Critic Update:** Minimize $J_Q$ for $Q_{\theta_1}$ and $Q_{\theta_2}$ (Eq.8)
8:     **Value Function Update:** Minimize $J_V$ (Eq.9)
9:     Update target networks $Q_{\bar{\theta}_1}$, $Q_{\bar{\theta}_2}$, and $V_{\bar{\psi}}$ via Polyak averaging
10:     **Compute Advantage:** $A(u_{i,k'}, a_{i,k'})$ (Eq.10)
11:     **Actor Update:** Minimize $J_\phi$ (Eq.11)
12:     Increment $t \leftarrow t + 1$
13: **end while**
14: **return** Appraisal Mechanism $\pi_\phi$

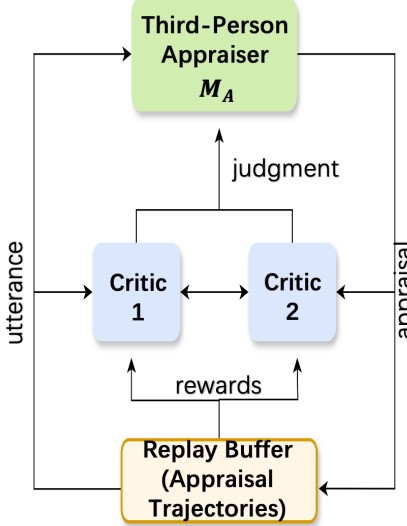

Figure 3: Diagram of meta-evaluation process

**Critic Model**: The critic evaluates the appraisals generated by $M_A$ and provides value estimates to guide the refinement of $M_A$'s policy. We train three Multi-Layer Perceptrons (MLPs)(Taud & Mas, 2018): two critics representing utterance-level Q-functions, $Q_{\theta_1}(u_{i,k'}, a_{i,k'})$ and $Q_{\theta_2}(u_{i,k'}, a_{i,k'})$, where $u_{i,k'}$ and $a_{i,k'}$ are sampled from $\mathcal{D}$. The double critic architecture is employed to reduce overestimation bias. Additionally, we have an MLP for the utterance-level value function $V_\psi(u_{i,k'})$. In this framework, $k'$ represents the iteration index in $\mathcal{D}$ for the $i$-th utterance. It ranges from $k' = 0$ (initial appraisal) up to $k' = K_i$, where $K_i$ is the total number of iterations for $i$-th utterance.

Target networks $Q_{\bar{\theta}_1}$ and $Q_{\bar{\theta}_2}$, and $V_{\bar{\psi}}$ are delayed copies of the respective models, updated via Polyak averaging (Polyak & Juditsky, 1992). The parameters $\theta_1$, $\theta_2$, and $\psi$ are the trainable parameters of the MLPs, while the target network parameters $\bar{\theta}_1$, $\bar{\theta}_2$, and $\bar{\psi}$ are updated using the moving averages of $\theta_1$, $\theta_2$, and $\psi$, respectively.

The Q-functions are trained by minimizing the Bellman error using targets derived from $V_{\bar{\psi}}$. The value function $V_\psi$ is trained to approximate the expected value of $Q_{\bar{\theta}_1}$ and $Q_{\bar{\theta}_2}$:

$$r_{i,k'} = \alpha r_{i,k'}^{\text{actor}} + \beta r_{i,k'}^{\text{critic}} \tag{5}$$

$$J_Q(\theta_j) = \mathbb{E}_{(u_{i,k'}, a_{i,k'}, r_{i,k'}) \sim \mathcal{D}}\left[\left(Q_{\theta_j}(u_{i,k'}, a_{i,k'}) - \left(r_{i,k'} + \gamma V_{\bar{\psi}}(u_{i,k'})\right)\right)^2\right], \quad j = 1, 2 \tag{6}$$

$$J_V(\psi) = \mathbb{E}_{(u_{i,k'}, a_{i,k'+1}) \sim \mathcal{D}}\left[\left(V_\psi(u_{i,k'}) - Q_{\bar{\theta}_1}(u_{i,k'}, a_{i,k'+1})\right)^2 + \left(V_\psi(u_{i,k'}) - Q_{\bar{\theta}_2}(u_{i,k'}, a_{i,k'+1})\right)^2\right] \tag{7}$$

where $\alpha$ and $\beta$ are weighting coefficients, and $\gamma$ is the discount factor. For terminal steps (i.e., when the process reaches its final step) where $k' = K_i$, we set $a_{i,k'+1} = a_{i,k'}$.

**Actor Model**: We train the third-person appraisal $M_A$ using an offline policy gradient approach, utilizing advantage values derived from the minimum of the two Q-values from the critic model. The advantage function is calculated as:

$$A(u_{i,k'}, a_{i,k'}) = \min\left(Q_{\theta_1}(u_{i,k'}, a_{i,k'}), Q_{\theta_2}(u_{i,k'}, a_{i,k'})\right) - V_\psi(u_{i,k'}) \qquad (8)$$

These advantage values guide the $M_A$ in refining its appraisal generation mechanism, leading to more accurate emotional appraisals. The policy gradient update is performed by minimizing:

$$J_\phi(\pi) = -\mathbb{E}_{(u_{i,k'}, a_{i,k'}) \sim \mathcal{D}}\left[A(u_{i,k'}, a_{i,k'}) \log \pi_\phi(a_{i,k'}|u_{i,k'})\right] \qquad (9)$$

where $\phi$ represents the trainable parameters of $M_A$.

## 3    EXPERIMENTS & RESULTS

In this section, we present five major experiments designed to evaluate the performance of our proposed model. The experiments are structured as follows: (1) a comparative analysis against LLM-based models; (2) an ablation study assessing the impact of appraisal knowledge dataset quality on the model's reasoning performance; (3) a comparative analysis of two verbal RL-based strategies for evaluating the effectiveness of the self-evaluation phase; (4) an ablation study assessing the meta-evaluation phase; and (5) a qualitative analysis of the model's appraisal performance on the DailyDialog dataset.

**Baselines:** For comparison, we use the instruction-tuned LLaMA3.1-8B-Instruct, Gemma1.1-7B-Instruct, and Mistral-7B-Instruct-v0.3 as baseline models.

**Evaluation Metrics:** We use accuracy (Acc.) and Weighted-F1 as our performance metrics for both IEMOCAP and DailyDialog datasets.

**Implementation Details:** We set the fixed window length, $l$, to 5. We utilize GPT-4 for $M_X$. For fine-tuning the third-person appraisal $M_A$, we utilize the LLaMA-3.1-8B-Instruct LLM. In the self-evaluation phase, the reflective cycle is set to 2 iterations. During the meta-evaluation phase, each of the double critic models is implemented as a 3-layer MLP, while the value model is implemented as a 2-layer MLP, with their embeddings initialized using pre-trained RoBERTa (Liu, 2019). Both the actor and critic models are trained using the Adam optimizer (Kingma & Ba, 2014) with the same learning rate of $1 \times 10^{-5}$. Training is conducted over 10 epochs. The constant coefficients $\alpha$ and $\beta$ are set to 0.9 and 0.45, respectively. The $M_A$ model is trained using 4-bit quantized low-rank adapters (LoRA) (Hu et al., 2021b), with $r = 16$. During inference (the test mode), the model's temperature is set to 0.8.

The dataset information is provided in the appendix (see Appendix A).

### 3.1    MAIN RESULTS

To demonstrate the effectiveness of our third-person appraisal agent, we benchmark it against instruction-tuned LLM baselines. We select the first 600 utterances from the IEMOCAP training dataset, generating 1,179 appraisal trajectories during the self-evaluation phase. These trajectories are then used to train the agent in the meta-evaluation phase. Finally, the fine-tuned agent is evaluated on the IEMOCAP test set, which comprises 1,623 utterances.

Table 1: Performance comparisons in accuracy and Weighted-F1 of our model against baselines on the IEMOCAP test set, reorganized by model types

| Model | Methods | Acc. | Weighted-F1 |
|---|---|---|---|
| Mistral | [1] Mistral-7B-Instruct-v0.3 (original) | 41.40 | 40.79 |
|  | [2] Mistral-7B-Instruct-v0.3 (ours) | 46.94 | 45.09 |
| Gemma | [3] Gemma1.1-7B-Instruct (original) | 42.62 | 43.64 |
|  | [4] Gemma1.1-7B-Instruct (ours) | 45.64 | 44.64 |
| LLAMA | [5] LLAMA-3.1-8B-Instruct (original) | 42.75 | 39.90 |
|  | [6] LLAMA-3.1-8B-Instruct (causal prompt) | 38.63 | 37.13 |
|  | [7] Ours (fine-tuned) | **50.96** | **51.33** |

Table 1 shows that the agent achieved the best performance on both prediction accuracy and weighted-f1 score by learning from a small number of samples, further validating the effectiveness of our approach.[6] uses the causal prompt (see Appendix B) fine-tuning method from (Team, 2024), guiding the LLAMA-3.1-8B-Instruct to identify emotion triggers and use those triggers to infer emotions. However, this method reduces performance by 4.12% compared to [5], likely due to the model's difficulty in understanding the causal relationship between emotional triggers and the speaker's emotional responses.

We also experiment with training 7B instruction-tuned LLMs exclusively during the meta-evaluation phase to assess their performance. The results show that our method achieves strong performance across all the baseline models listed in the table [2,4,7]. Specifically, our model[7] shows a significant improvement of 8.21% over the original LLAMA-3.1-8B-Instruct model[5], which uses a general prompt to infer emotions based only on the information from the provided dialogue context.

## 3.2 ABLATION STUDY ON APPRAISAL KNOWLEDGE DATA GENERATION

This ablation study examines how modifications to appraisal knowledge generation affect the third-person appraisal agent's learning performance. We focus on three key factors: 1) removal of emotion labels, 2) removal of speaker information, and 3) replacement of the AppraisalKnowledge prompt with a general summary prompt. Using five input configurations on the same 600 utterances from the IEMOCAP training dataset, we generate different appraisal knowledge with GPT-4 (see Table 2), train each agent using these five datasets, and evaluate their performance on the IEMOCAP test set. We find that configurations lacking both emotion labels and speaker information exhibited minimal performance changes compared to those utilizing the AppraisalKnowledge prompt, highlighting its essential role in generating quality knowledge. Overall, configurations omitting all three factors led to a 9.74% decrease in accuracy and an 8.55% drop in the weighted F1 score, indicating their importance in generating better appraisal knowledge.

Table 2: Performance of agents trained on datasets generated from five different input configurations, evaluated on the IEMOCAP test set. The table highlights how the inclusion or exclusion of true emotion labels, speaker information, and the AppraisalKnowledge prompt influence the agents' performance in terms of accuracy (Acc.) and weighted F1 score.

| Data Input Configurations | | | | |
|:---:|:---:|:---:|:---:|:---:|
| **True Label** | **Speaker Info.** | **AppraisalKnowledge Prompt** | **Acc.** | **Weighted-F1** |
| ✓ | ✓ | ✗ | 44.92 | 43.14 |
| ✗ | ✗ | ✓ | 45.44 | 45.71 |
| ✗ | ✓ | ✓ | 47.63 | 47.92 |
| ✗ | ✗ | ✗ | 41.59 | 42.78 |
| ✓ | ✓ | ✓ | **50.96** | **51.33** |

**Reflexion vs. Counterfactual Reasoning:**
In Figure 4, we show the percentage change in correct predictions after each reflective iteration, using the no self-evaluation baseline for reference.
We observe that Reflexion yields moderate improvements, whereas counterfactual reasoning leads to a nearly 17.65% increase after the third iteration. This suggests that counterfactual reasoning outperforms Reflexion in enhancing correct predictions during self-evaluation. One possible explanation is that Reflexion offers limited improvement in emotional reasoning, as it only allows the agent to reflect on errors without providing specific guidance for adjustments.

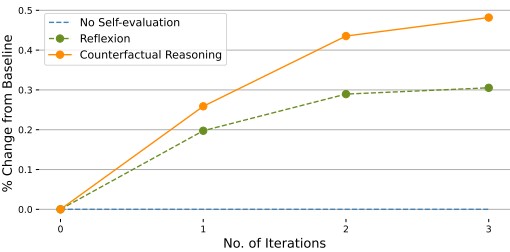

Figure 4: Percentage change of correct samples over the self-evaluation phase, relative to the baseline values from the No Self-evaluation phase.

### 3.3 ANALYSIS OF SELF-EVALUATION PHASE

To demonstrate the effectiveness of the counterfactual reasoning strategy, we conduct a comparative experiment against the Reflexion-based method Shinn et al. (2024); Koa et al. (2024). We select 100 utterances from the IEMOCAP training dataset and apply both strategies during self-evaluation. See the details in Figure 4.

### 3.4 ABLATION STUDY OF META-EVALUATION PHASE

In the meta-evaluation phase, we conduct an ablation study on different variants of the model. For each variant, we remove one specific component: 1) no actor rewards during RL, 2) no critic rewards during RL, 3) no reflective actor-critic RL, and 4) no inference instruction, where the agent is instruction-tuned to predict emotion states without the InferenceInstruction Prompt. We conduct comparisons using the IEMOCAP test set.

Table 3 demonstrates that incorporating both actor and critic rewards enhances the agent's self-appraisal capabilities. Despite being trained on a small dataset, the reflective actor-critic RL approach achieves a 1.61% increase in accuracy, indicating that this RL strategy can further enhance the agent's ability to generate accurate appraisals. Conversely, removing the InferenceInstruction prompt results in a significant 8.38% drop in accuracy, indicating that the appraisal-based chain-of-thought instruction plays a crucial role in guiding the model's reasoning process (Chung et al., 2024).

Table 3: Ablation study on meta-evaluation phase.

| Methods | Acc. | Weighted-F1 |
|---|---|---|
| w/o Actor Rewards | 49.23 | 49.56 |
| w/o Critic Rewards | 49.47 | 49.95 |
| w/o Reflective Actor-Critic RL | 49.35 | 49.51 |
| w/o InferenceInstruction | 42.58 | 43.32 |
| Ours | **50.96** | **51.33** |

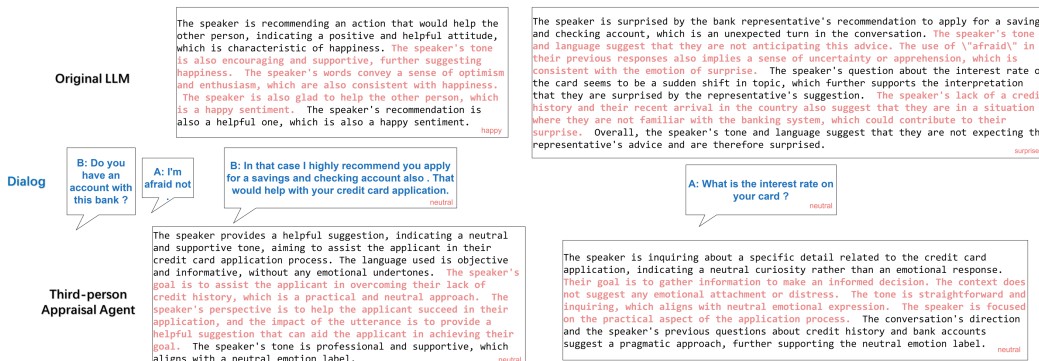

Figure 5: The figure depicts the reasoning processes of the agent and the original LLM, respectively, for a dialogue excerpt from the DailyDialog test set, with key sentences highlighted in red to indicate their respective reasoning steps.

### 3.5 THE PERFORMANCE OF APPRAISALS

To demonstrate the reasoning capabilities of our agent, we randomly select 1,555 samples from the DailyDialog dataset to evaluate its generalization performance. Our third-person appraisal agent outperforms the original LLM integrated into LLAMA-3.1-8B-Instruct. As shown in Table 4, our agent achieves a 14.93% higher accuracy compared to the original LLM, indicating that enhancing the reasoning capabilities can significantly improve the LLMs' accuracy in predicting human emotions.

Additionally, we compare the appraisals generated by the original LLM with those generated by our third-person appraisal agent. Two key improvements are observed in this experiment. First, our agent demonstrates advanced reasoning by evaluating the speaker's mental states—such as attitudes, goals, desires, and expectations—using contextual information. For example, Figure 5 demonstrates the comparative reasoning performance of our agent versus the original LLM on a conversation excerpt. Our agent effectively identifies underlying causes, such as the speaker's motivations and intentions, going beyond basic emotional triggers. In contrast, the original LLM primarily focuses on identifying emotion triggers and provides limited reasoning based on surface-level cues and sentiments.

Table 4: Performance comparison between the original LLM and our third-person appraisal agent on the DailyDialog dataset

| Methods | Acc. | Weighted-F1 |
|---------|------|-------------|
| original | 41.72 | 50.13 |
| ours | **56.65** | **63.62** |

Table 5: Comparison of appraisal quality between the original LLM and our third-person appraisal agent

| Metric | Original | **Ours** |
|--------|----------|----------|
| Sentiment Awareness | 4.71 | **4.99** |
| Contextual Understanding | 4.58 | **4.68** |
| Sensitivity to Emotional Causes | 4.43 | **4.58** |
| Emotional Dynamics Responsiveness | 4.19 | **4.38** |
| Motivational Understanding | 4.41 | **5.13** |
| Clarity and Coherence Assessment | 4.60 | **4.77** |

Our agent shows an improved ability to generate qualitative appraisals, which is a challenging task for LLMs as it requires understanding how conversational utterances influence emotions. To assess our agent's appraisal quality compared to the original LLM, we develop a set of appraisal quality metrics and use GPT-4 to rate each appraisal on a scale of 1 to 6 using the same DailyDialog test set. The average scores for each metric are shown in Table 5, with detailed explanations provided in Appendix C. Based on these results, we make the following observations:

- The original LLM achieves the highest sentiment awareness score across all of its metrics, highlighting its strong emphasis on sentiment analysis in its reasoning process.

- Both models perform well on clarity and coherence, indicating their ability to generate well-structured appraisals.

- Our model excels in motivational understanding, demonstrating a strong focus on identifying motivations when analyzing emotions.

- Key metrics for evaluating the model's reasoning performance include sentiment awareness, contextual understanding, responsiveness to emotional dynamics, and comprehension of motivations. The table shows that our model outperforms the baseline model in all four metrics, demonstrating its superior reasoning capabilities for the ERC task.

## 4 CONCLUSION

We integrate cognitive appraisal theory with a novel learning framework to train an agent capable of performing emotional reasoning from a third-person perspective. This approach allows the agent to continually refine its emotion reasoning abilities, even with a limited amount of data. Our approach advances the development of explainable AI by training the agent to perform emotion reasoning in a way that more closely aligns with human emotional understanding.

A key limitation of our work is the inherent difficulty LLMs face in interpreting complex emotional transitions. For example, understanding how an extremely positive emotion like 'happiness' can shift into an extremely negative one like 'sadness' remains a major challenge. Addressing these limitations will be a primary focus of our future research as we aim to further improve the agent's ability to comprehend and reason through complex emotion shifts.

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

## A  DATASET

The Third-person Appraisal Agent is evaluated on the IEMOCAP benchmark dataset (Busso et al., 2008), which consists of conversational utterances paired with corresponding emotion labels. To further demonstrate the generalization capability of our framework, we evaluated it on the DailyDialog dataset (Li et al., 2017), which contains previously unseen emotion labels.

IEMOCAP Busso et al. (2008) comprises dyadic conversations between ten speakers, with the training set derived from the first eight participants. Each video captures a dyadic dialogue, divided into utterances annotated with six emotions: happy, sad, neutral, angry, excited, and frustrated.

DailyDialog Li et al. (2017) covers various everyday topics, mirroring natural human conversation. Each utterance is annotated with emotional categories and dialogue acts, including seven emotions: angry, disgusted, fearful, joyful, neutral, sad, and surprised.

| Dataset | Dialogues | | | Utterances | | | Avg. Length | Classes |
|---------|-----|-----|------|-------|-------|-------|--------|---------|
| | train | val | test | train | val | test | | |
| IEMOCAP | 108 | 12 | 31 | 5,810 | — | 1,623 | 47 | 6 |
| DailyDialog | 11,118 | 1,000 | 1,000 | 87,832 | 7,912 | 7,863 | 72 | 7 |

Table 6: The statistics of two datasets.

# B  FULL PROMPTS AND THEIR RESPONSES

```
AppraisalKnowledge_PROMPT = """
Given a dialogue context {dialog} and a true emotion label {emotion},
    analyze the target utterance of {utterance} to generate its
    appraisal knwowlegde. Follow three steps:

1. Identify the key elements of the situation from the given dialogue.
2. Evaluate relation to speaker's goals, intentions, desires, or
    expectations on the target utterance.
3. Determine the relevance and potential impact of the situation on the
    speaker, focusing specifically on the target utterance.

Your response:
"""

Here are several examples of applying AppraisalKnowledge Prompt template.
##Example 1
utterance: F: What? I'm getting an ID. This is why I'm here. My wallet
    was stolen.
dialog: M: Okay. But I didn't tell you to get in this line if you are
    filling out this particular form. F: Well what's the problem? Let me
    change it. M: This form is a Z.X.four. M: You can't-- This is not
    the line for Z.X.four. If you're going to fill out the Z.X.four, you
    need to have a different form of ID. F: What? I'm getting an ID.
    This is why I'm here. My wallet was stolen.
emotion: frustrated

appraisal knowledge:
situation: In response to being told she needs different ID, the female
    speaker explains her predicament of needing an ID because her wallet
    was stolen, which is why she is there.
speaker's perspective: Her exclamation and explanation aim to convey her
    frustrating situation and the necessity of her visit, seeking
    understanding or assistance in a difficult circumstance.
impact: Her disclosure introduces a personal crisis element into the
    interaction, which may elicit sympathy or prompt a more helpful
    response from the institution to accommodate her needs despite the
    procedural hiccup.

##Example 2
utterance: M: I know. All right, all right. All right. Okay. Calm
    yourself. What does that mean, me above all?
dialog: M: BREATHING M: Calm yourself. F: Just believe with me, Joe.
    Only last week a man came back in Detroit missing longer than Larry.
    Believe with me. You, above all, have got to believe. Just believe.
    M: Okay. Calm yourself. M: I know. All right, all right. All right.
    Okay. Calm yourself. What does that mean, me above all?
emotion: frustration

appraisal knowledge:
situation: The male speaker responds with several acknowledgments, but
    then questions the female's statement about his unique role in her
    belief.
```

speaker's perspective: The male speaker is showing signs of frustration
    or resignation by repeating phrases, and then challenges her
    assertion that he holds a special responsibility in maintaining her
    hope.
impact: This question shifts the focus from her plea to his own role in
    the situation, potentially opening up a deeper discussion about
    their relationship and responsibilities, but also risking more
    tension if she interprets his words as resistance.

________________________________________________

AppraisalGenerator_PROMPT = """
Instruction: Deduce the emotion behind the given utterance, using ONLY
    the provided dialog and premises. Do not make assumptions beyond the
    given information. Respond using ONLY the following format:

Emotion Label: [choose one from: happy, sad, neutral, angry, excited,
    frustrated]
Appraisal: [Your reasoning in 2-3 short sentences]

Utterance: {utterance}
Dialog context: {dialog}
Premises: {appraisal knowledge}

Your Response:
"""

Here is an example of applying AppraisalGenerator Prompt template.

utterance: F: Well what's the problem? Let me change it.

dialog: M: Let me see them. F: Is there a problem? M: Who told you to
    get in this line? M: Okay. But I didn't tell you to get in this line
    if you are filling out this particular form. F: Well what's the
    problem? Let me change it.

appraisal Knowledge:
    situation: After being informed she's in the wrong line, the female
        speaker offers to correct the situation by changing lines,
        showing readiness to comply.
    speaker's perspective: Her response aims to quickly resolve the
        issue and continue her task without further delay, demonstrating
        a cooperative stance.
    impact: Her willingness to adapt could de-escalate potential tension
        and facilitate a smoother continuation of her task within the
        correct procedural framework.

prediction: neutral
appraisal: The speaker's response suggests a cooperative and
    problem-solving attitude, indicating a neutral emotional tone. She
    is focused on resolving the issue and continuing her task without
    delay, showing no apparent frustration or annoyance. Her offer to
    change lines demonstrates a neutral, solution-focused approach. The
    speaker's tone is matter-of-fact and practical, indicating a neutral
    emotional state. Her response does not convey any strong emotions or
    reactions to the situation. Her willingness to adapt and continue
    her task indicates a neutral, cooperative stance. The speaker's goal
    is to resolve the issue and proceed, without expressing any strong
    emotions. Her response is direct and solution-focused, indicating a
    neutral emotional tone.

________________________________________________

```
CounterfactualReasoning_PROMPT = """
Instruction:
What if the speaker's emotional response wasn't {previous_label}, but
    instead displayed a different emotion?

Steps:
1. Premises: Carefully re-examine each {appraisal knowledge}
2. Utterance: Identify key emotional indicators in {utterance}
3. Counterfactual Emotion:
   a. Predict an emotion that contradicts the apparent emotion in the
       utterance
   b. Analyze how this contradictory emotion could fit the situation
   c. Explore the implications if the speaker were feeling this
       contradictory emotion

Response Format:
Emotion Label: [choose one from: happy, sad, neutral, angry, excited,
    frustrated]
Appraisal: [Your reasoning in 2-3 short sentences]

Your Response:
"""

Here is an example of applying the Counterfactual Reasoning Prompt
    template.

utterance: F: That would - I would appreciate that. Yeah.
appraisal knowledge:
situation: The female speaker affirms her desire to speak with a
    supervisor, politely expressing appreciation for the action being
    taken despite the previous tension.
speaker's perspective: Her response shows an effort to maintain decorum
    and cooperation, hoping that the involvement of a supervisor will
    lead to a resolution of her problem.
impact: This polite insistence on escalation could facilitate a more
    careful review of her situation, potentially leading to a more
    empathetic and flexible handling of her case.

###Previous trial:
previous label: neutral
previous appraisal: The female speaker uses polite language, such as \"I
    would appreciate that\" and \"Yeah\", which suggests a neutral tone,
    indicating she is seeking a resolution without expressing strong
    emotions. Her goal is to escalate the situation to a supervisor,
    implying a focus on finding a solution rather than expressing
    negative emotions. The polite tone and language used maintain a
    level of professionalism and decorum, further indicating a neutral
    emotion. The speaker's words and tone suggest she is trying to
    navigate the situation effectively rather than expressing a strong
    emotional state. Her response shows a neutral, matter-of-fact
    approach, which is in line with her goal of finding a solution.
    Overall, her tone and language suggest she is neutral and focused on
    finding a resolution.

###After counterfactual reasoning:

prediction: frustrated
adjusted appraisal:
If the speaker were feeling frustrated, it would suggest that she's not
    just seeking a resolution, but is also experiencing a sense of
    exasperation or annoyance with the situation. This could imply that
    she's been dealing with the issue for a while and is getting tired
    of the bureaucratic process. The frustrated tone would also indicate
```

that she's not just neutral, but is actively seeking a solution to
alleviate her frustration. This would change the dynamics of the
interaction, as she might be more direct or forceful in her approach
to resolve the issue. The polite language would still be present,
but it would be a facade for her underlying frustration. This would
also imply that she's not just focused on finding a solution, but is
also motivated by a desire to address the root cause of her
frustration. The frustrated tone would add a layer of complexity to
her goal of escalating the situation to a supervisor, as she might
be more likely to take a confrontational approach to get her point
across. Overall, a frustrated tone would change the tone and
dynamics of the interaction, making it more intense and potentially
confrontational. The frustrated tone would also suggest that she's
not just seeking a resolution, but is also motivated by a desire to
address the root cause of her frustration. The polite language would
still be present, but it would be a facade for her underlying
frustration.

————————————————————————————————————————————————

```
Evaluation_PROMPT = f'''
Given the range of the class of {emotion} in the Circumplex Model of
    Affect, do the valence score of {valence} and the arousal score of
    {arousal} together fit within this range?

Answer only 'yes' or 'no'.
'''
```

————————————————————————————————————————————————

```
InferenceInstruction_Prompt = """
Analyze the given utterance within its dialogue context. Provide a
    concise appraisal and predict an emotion label in the following
    format:

Situation: [Brief context description]
Speaker's perspective: [Speaker's goals or intentions]
Impact: [The impact of the utterance on the conversation]

Keep each section to 1-2 sentences. Base your analysis solely on the
    provided dialogue.
Dialogue context: {dialogue}
Utterance to analyze: {utterance}

Response Format:
Emotion Label: [choose one from: happy, sad, neutral, angry, excited,
    frustrated]
Explanation: [Brief appraisal explaining the chosen emotion label]

Response:
"""
```

```
causal_prompt = """
You are an expert in emotion classification and emotion cause
    recognition. The following is a conversation that involves several
    speakers. Analyze each utterance within its context and identify the
    potential cause of the emotion expressed in the utterance before
    predicting the emotion label.

Dialogue context: {dialogue}
Utterance to analyze: {utterance}

Response Format:
Emotion Label: [choose one from: happy, sad, neutral, angry, excited,
    frustrated]
```

```
Explanation: [Chosen emotion label based on the identified cause of the
    emotion]

Response:
"""
```

## C   EVALUATION OF APPRAISAL QUALITY

The metrics below assess the quality of emotional reasoning by evaluating the model's generated appraisals. The following descriptions detail the metrics, curated with the assistance of ChatGPT. Given the novelty of this field, research on evaluating emotional appraisals is limited.

• Sentiment Awareness

Definition: Measures the model's ability to recognize and accurately interpret the emotional tone and sentiment in communication, reflecting the speaker's feelings and attitudes.

**Evaluation Criteria:**

Does the appraisal effectively identify and differentiate between various emotional tones?

Does the appraisal consider the intensity of the expressed emotions?

• Contextual Understanding

Definition: Assesses the model's capacity to comprehend and integrate contextual cues when interpreting emotions.

**Evaluation Criteria:** Does the appraisal consider contextual cues that influence emotions?

• Sensitivity to Emotional Causes

Definition: Evaluate the model's ability to identify and understand the underlying causes of expressed emotions.

**Evaluation Criteria:**

Does the appraisal accurately identify and articulate the reasons or events that led to the expressed emotions?

• Emotional Dynamics Responsiveness

Definition: Assesses the model's capability to detect and respond to changes in emotional states over time.

**Evaluation Criteria:**

Does the appraisal effectively track and reflect changes in emotions throughout the conversation?

• Motivational Understanding

Definition: Measures the model's ability to recognize motivations of individuals behind their emotional expressions.

**Evaluation Criteria:**

Does the appraisal identify the speaker's motivations or goals behind their emotional state?

Does the appraisal reflect an understanding of how the speaker's emotional expressions relate to their desires or anticipated outcomes?

• Clarity and Coherence Assessment

Definition: Assess the clarity and coherence of the generated appraisals.

**Evaluation Criteria:**

Is the appraisal clear and easy to understand?

Does the interpretation flow coherently, linking emotional insights to contextual information?

