# OpenReview forum: "A Third-Person Appraisal Agent: Learning to Reason About Emotions in Conversational Contexts"
_ICLR.cc/2025/Conference — ICLR 2025 Conference Withdrawn Submission_

### Official Review · Reviewer_eyXK · 2024-10-22

**Soundness:** 2
**Presentation:** 2
**Contribution:** 2
**Rating:** 5
**Confidence:** 4

**Summary:**

In the area of Emotion Recognition in Conversation (ERC), this paper proposed a learning framework based on cognitive appraisal theory, utilizing an agent powered by LLMs to learn emotion reasoning from a third-person perspective. Experimental results on two benchmark datasets of ERC, IEMOCAP and Dailydialog, validate that the proposed method outperforms some LLM baselines.

**Strengths:**

1. This paper describes the motivation, the proposed methodology, and the experimental in details, which is easy to follow.
2. Motivation for emotional reasoning from a third-person perspective is interesting and promising.
3. Experimental results show that the proposed third-person appraisal agent outperforms Mistral-7B-Instruct-v0.3, Gemma1.1-7B-Instruct, and LLAMA-3.1-8B-Instruct on the IEMOCAP test set.
4. A large number of ablation experiments were conducted to verify the effectiveness of appraisal knowledge generation, self-evaluation phase, meta-evaluation phase. In addition, experiments were conducted on DailyDialog to demonstrate generalization.

**Weaknesses:**

1. The representations of Fig. 1, Fig.2, and Fig.5 should be optimized. For example, the font size of the emotion labels in Figure 1, as well as the alignment of the text boxes; the module sizes and framework alignment in Figure 2; and the alignment of the text boxes in Figure 5, among others.
2. Improve the accuracy of emotion recognition by employing large language models to emotional reasoning is promising, but the experimental results did not highlight the advantages of large language models for emotional reasoning and comprehension. Previous ERC work [1] [2] [3] [4] have significantly higher emotion recognition accuracy than the proposed third-person appraisal agent, even without emotional reasoning.
3. The experiments in this paper are conducted only on a subset of the IEMOCAP and DailyDialog datasets. Such a small amount of data is insufficient to demonstrate the effectiveness of the proposed method. More datasets should be considered as experimental benchmarks, e.g., MELD [5], M3ED [6], EmoryNLP [7], etc.
4. While ERC in the textual modality is certainly relevant, generalization to multimodal scenarios (visual, audio, and textual) is even more relevant and interesting [1] [2] [4].

[1] Hu et al. UniMSE: Towards Unified Multimodal Sentiment Analysis and Emotion Recognition. 2022.

[2] Li et al. EmoCaps: Emotion Capsule based Model for Conversational Emotion Recognition. 2022.

[3] Lei et al. InstructERC: Reforming Emotion Recognition in Conversation with a Retrieval Multi-task LLMs Framework. 2023.

[4] Hu et al. UniMEEC: Towards Unified Multimodal Emotion Recognition and Emotion Cause. 2024.

[5] Poria et al. MELD: A Multimodal Multi-Party Dataset for Emotion Recognition in Conversation. 2019.

[6] Zhao et al. M3ED: Multi-modal Multi-scene Multi-label Emotional Dialogue Database. 2022.

[7] Zahiri et al. Emotion Detection on TV Show Transcripts with Sequence-based Convolutional Neural Networks. 2017.

**Questions:**

1. Why set the fixed window length to 5, reflective cycle to 2? Is there a priori knowledge or is it based on empirical settings?
2. How fast is agent's emotional reasoning and emotion recognition? Is it comparable to previous ERC models?

---

> ### Author Response · Authors · 2024-11-20
> **Response to reviewer eyXK**
>
> 1.	The representations of Fig. 1, Fig.2, and Fig.5 should be optimized. For example, the font size of the emotion labels in Figure 1, as well as the alignment of the text boxes; the module sizes and framework alignment in Figure 2; and the alignment of the text boxes in Figure 5, among others.
>
> **Answer:** Thank you for your feedback; we have revised the figures to improve readability and alignment (we will update our revised version later.)
>
> 2.	Improve the accuracy of emotion recognition by employing large language models to emotional reasoning is promising, but the experimental results did not highlight the advantages of large language models for emotional reasoning and comprehension. Previous ERC work [1] [2] [3] [4] have significantly higher emotion recognition accuracy than the proposed third-person appraisal agent, even without emotional reasoning.
>
> **Answer:** Thank you for your insightful feedback. We acknowledge that our experimental results did not surpass the emotion recognition accuracy of previous ERC work. However, several factors contribute to this outcome:
>
> •	Inherent Capabilities of Seed LLMs: Research [1-2] indicates that the application of knowledge to specific tasks is constrained by the inherent capabilities of the base LLMs. The seed models may lack the specialized training necessary for emotional reasoning, which can limit their performance in emotion recognition tasks.
>
> •	Computational Constraints: Due to limited computational resources, we were unable to apply our methods to larger models, such as the 13B or 72B parameter variants. Fine-tuning these larger models requires significantly more GPU resources, which were beyond our current capacity. Nonetheless, our results with the 7B and 8B parameter baseline LLMs (see Table 1) demonstrated improved performance, validating the effectiveness of our approach. We believe that applying our method to larger models could further enhance their reasoning capabilities and accuracy.
>
> Our primary objective is to enhance the emotional reasoning capabilities of LLMs rather than solely maximizing emotion recognition accuracy. While previous ERC models achieve high accuracy, they often lack the ability to provide interpretable reasoning behind their predictions. To address this limitation, we reformulated the emotion recognition task as a Q&A problem, requiring the model to generate answers accompanied by a detailed reasoning process.
>
> [1] Lu, K., Yu, B., Zhou, C., & Zhou, J. (2024). Large Language Models are Superpositions of All Characters: Attaining Arbitrary Role-play via Self-Alignment. In Proceedings of the 62nd Annual Meeting of the Association for Computational Linguistics.
>
> [2] Zhao, W. X., Zhou, K., Li, J., Tang, T., Wang, X., Hou, Y., ... & Wen, J. R. (2023). A survey of large language models. arXiv preprint arXiv:2303.18223.
>
> 3.	The experiments in this paper are conducted only on a subset of the IEMOCAP and DailyDialog datasets. Such a small amount of data is insufficient to demonstrate the effectiveness of the proposed method. More datasets should be considered as experimental benchmarks, e.g., MELD [5], M3ED [6], EmoryNLP [7], etc.
>
> **Answer:**  We appreciate the suggestion and have taken steps to address it. To further validate the effectiveness of our proposed method, we conducted additional experiments using the EmoryNLP dataset, which includes 1,328 test samples labeled into seven emotion classes: 'neutral,' 'peaceful,' 'joyful,' 'mad,' 'sad,' 'powerful,' and 'scared.' Despite these labels differing significantly from those in the IEMOCAP dataset, which classifies emotions as 'happy,' 'sad,' 'neutral,' 'angry,' 'excited,' and 'frustrated,' our method was rigorously evaluated against the original LLM integrated into LLAMA-3.1-8B-Instruct. As shown in the table below, our approach achieves a 6.32% higher accuracy than the original LLM.
>
> Table: Performance comparison between the original LLM and our third-person appraisal agent on the EmoryNLP dataset:
>
> Methods   | Accuracy | Weighted-F1
>
>   Original    | 23.36    | 20.66
>
>    Ours      | 29.68    | 27.67
>
>
> 4.	While ERC in the textual modality is certainly relevant, generalization to multimodal scenarios (visual, audio, and textual) is even more relevant and interesting [1] [2] [4].
>
> **Answer:**  While emotion reasoning in the ERC field has received limited research attention, our work provides a potential solution to advance this area. Text-based ERC faces significant challenges, particularly in enabling LLMs to capture and interpret intricacies in conversational complexity—an issue that is highly relevant in real-world applications such as customer service, mental health assessments, and human-computer interactions. In future work, we will consider incorporating multimodal data (visual, audio, and textual) to further enhance the emotional reasoning capabilities of LLMs.

---

> > ### Comment · Reviewer_eyXK · 2024-11-24
> > **Response to Authors**
> >
> > Thank you for your response, some of my concerns have been addressed. However, My primary concern lies with the motivation of the proposed method, which is to leverage emotion reasoning to enhance the model's performance in emotion recognition. However, the experimental results presented in the paper do not demonstrate a clear advantage over the existing state-of-the-art (SOTA) methods. In fact, there appears to be a significant performance gap compared to SOTA, which raises questions about the effectiveness of the proposed approach.
> >
> > Additionally, the authors mention that due to limited computational resources, they were unable to conduct experiments with larger-scale backbones to demonstrate the effectiveness of their approach. However, it is important to note that existing SOTA methods typically utilize models with only a few hundred million parameters, which is significantly smaller than the 7B and 8B parameter baselines employed in this work. The fact that smaller models achieve better performance than the proposed approach diminishes the overall contribution and value of this study.

---

> > > ### Author Response · Authors · 2024-11-24
> > > **Response to reviewer eyXK**
> > >
> > > Thank you so much for your thoughtful feedback! We appreciate the opportunity to clarify our motivations and address the points you've raised.
> > >
> > > Our primary goal is to **bridge the gap in emotion reasoning within the ERC field**. While existing SOTA methods excel in emotion recognition by accurately predicting emotion labels, they often do so without providing insights into the underlying reasoning processes. This **limitation** restricts a deeper understanding of how models interpret and contextualize emotional states in conversations.
> > >
> > > Currently, **emotion reasoning performance has been evaluated primarily based on emotion labels, offering limited insights into the reasoning process**. However, in our paper, we introduce **an objective evaluation method to assess emotion reasoning performance more comprehensively**. This method emphasizes three key aspects: emotional comprehension, contextual understanding, and expressive coherence and clarity. Further details about this evaluation approach can be found in Section 3.5: The Performance of Appraisals, starting on line 479.
> > >
> > > Moreover, we would like to clarify that, while large language models (LLMs), such as GPT-4 or GPT-4o, have demonstrated extremely high performance in emotion recognition across various domain-specific tasks, **their capabilities in emotion reasoning remain limited**. Despite their success in identifying emotions, these models often struggle to provide coherent and contextually grounded reasoning about emotional states. **This gap highlights the need for focused efforts to improve the reasoning abilities of LLMs to enhance their understanding and interpretation of emotions in complex, real-world scenarios.**

---

> > > > ### Comment · Reviewer_eyXK · 2024-12-03
> > > > **Response to Authors**
> > > >
> > > > Thank you for your response. It is difficult to fully understand your statement in the rebuttal: "In our paper, we introduce an objective evaluation method to assess emotion reasoning performance more comprehensively." Emotion understanding varies from person to person, so how can an objective evaluation method for emotion reasoning be constructed in this field? Additionally, the experimental results only demonstrate that the proposed third-person appraisal agent outperforms the original LLM in emotion reasoning capabilities due to fine-tuning. However, it does not achieve the expected performance in emotion recognition compared to the current SOTA methods.
> > > >
> > > > Due to the concerns outlined above, I will maintain my current score.

---

> ### Author Response · Authors · 2024-11-20
> **Response to reviewer eyXK**
>
> Continue to the Questions:
>
> 1.	Why set the fixed window length to 5, reflective cycle to 2? Is there a priori knowledge or is it based on empirical settings?
>
> **Answer:** The fixed window length of 5 and reflective cycle of 2 were chosen based on empirical settings to optimize model performance.
>
> 2.	How fast is agent's emotional reasoning and emotion recognition? Is it comparable to previous ERC models?
>
> **Answer:**  After fine-tuning the LLM, during the inference phase, our model generates not only an emotion recognition decision but also a detailed interpretation of its reasoning process. This includes producing a chain of thought that explains how the model reached its conclusion. Consequently, our model incurs higher computational costs and longer processing times compared to previous ERC models that provide direct predictions without explicit reasoning.

---

### Official Review · Reviewer_3VTN · 2024-11-01

**Soundness:** 2
**Presentation:** 2
**Contribution:** 2
**Rating:** 3
**Confidence:** 4

**Summary:**

The authors propose a new framework to perform the task of ERC. They take inspiration from the cognitive appraisal theory and present a two step process for ERC -- In the first step, the authors generate appraisals using an LLM and improve these generated appraisals using verbal RL. In the next step, the trajectory of improvement along with dialogue context is passed on to a trainable LLM to learn emotions. The proposed method shows better results than the models authors compared their system with.

**Strengths:**

- The authors propose an interesting approach to the task of ERC. Emotion reasoning makes sense to mimic human emotion formation.
- The language of the paper is easy to follow.

**Weaknesses:**

- One major flaw is in how the authors generate the initial appraisals. The authors provide the true emotion labels in the prompt to generate the appraisals thus swaying the LLM in favor of generating the appraisals for that emotion. However, as mentioned in the cognitive appraisal theory, emotions *arise* from appraisals, which means that appraisals come before emotions, which is not the case for how the prompt is constructed. This then violates what the authors claim in Section 2.3 that "This LLM simulates human cognitive appraisal when reasoning about emotional states."
- It is unclear what is "appraisal knowledge xi" in section 2.3.1 or "appraisal context xi" in Algorithm 1.
- From Table 2, it is evident that the "AppraisalKnowledge Prompt" is the main contributor to the model performance. I feel like it might be due to the fact that that prompt contains the true emotion label, which in turn makes the appraisal knowledge contain that emotion label (as is the case in Figure 1. The appraisal contains "frustrated" and "angry" which are the emotion labels).
- The paper seems rushed with multiple typos and silly mistakes. For example:
   - Figure 3: "reply" --> "replay"
   - Table 6: "Table 1"? "three datasets" --> "two datasets"

**Questions:**

- It would be interesting to talk about the multi party nature of most of the real world conversations. And how can this multi-party phenomenon affects the proposed method.
- Speaker information in the instruction part would also be an interesting thing to see.
- Are the LLMs mentioned in Table 1 also finetuned?

---

> ### Author Response · Authors · 2024-11-20
> **Response to Reviewer 3VTN**
>
> •	One major flaw is in how the authors generate the initial appraisals. The authors provide the true emotion labels in the prompt to generate the appraisals thus swaying the LLM in favor of generating the appraisals for that emotion. However, as mentioned in the cognitive appraisal theory, emotions arise from appraisals, which means that appraisals come before emotions, which is not the case for how the prompt is constructed. This then violates what the authors claim in Section 2.3 that "This LLM simulates human cognitive appraisal when reasoning about emotional states."
>
> **Answer**: Thank you for your feedback and for highlighting this important aspect. We understand the concern regarding the order of appraisals and emotions in the context of cognitive appraisal theory, where emotions arise as a result of appraisals. We would like to address the potential misunderstanding related to the use of emotion labels in the initial prompt.
>
> Our work introduces a learning system specifically designed for fine-tuning a third-person appraisal LLM, referred to as M_A (see Figure 2 in our paper). **During the inference phase, the fine-tuned M_A operates independently by first generating appraisals based solely on the input context—without any emotion labels or appraisal knowledge—and then inferring emotions from these appraisals (as outlined in lines 199-201), aligning with our claim that "This LLM simulates human cognitive appraisal when reasoning about emotional states."**
>
> While our approach uses true emotion labels in the initial training phase, this step is strictly confined to the Appraisal-Knowledge Generation component. **Given the scarcity of emotion reasoning-related datasets in the ERC field, we leveraged emotion labels solely as informational anchors to guide the learning system in generating high-quality appraisal-related knowledge.** This is a necessary practical solution to address the lack of annotated emotion evaluation data, enabling our model to learn the complex relationships between contextual situations and speakers' intentions (as outlined in lines 155-158 of our paper).
>
> In the research field of causal reasoning for large language models (LLMs), many studies [1-4] use diverse methods such as synthetic data generation, knowledge graph assistance, real data annotation, and combined methodologies involving both synthetic and real-world data collection. These approaches help construct causal datasets that effectively evaluate and enhance the causal reasoning capabilities and performance of LLMs. Similarly, our use of emotion labels in the initial training phase is consistent with existing practices aimed at building appraisal-knowledge data to improve LLMs' reasoning processes.
>
> We hope this explanation addresses your concerns and clarifies how our approach aligns with Cognitive Appraisal Theory while leveraging emotion labels solely for initial knowledge generation purposes.
>
> [1] Rawal, A., Raglin, A., Wang, Q., & Tang, Z. Investigating Causal Reasoning in Large Language Models. In Causality and Large Models@ NeurIPS 2024.
>
> [2] Cai, H., Liu, S., & Song, R. (2023). Is Knowledge All Large Language Models Needed for Causal Reasoning?. arXiv preprint arXiv:2401.00139.
>
> [3] Kıcıman, E., Ness, R., Sharma, A., & Tan, C. (2023). Causal reasoning and large language models: Opening a new frontier for causality. arXiv preprint arXiv:2305.00050.
>
> [4] Liu, X., Xu, P., Wu, J., Yuan, J., Yang, Y., Zhou, Y., ... & Huang, F. (2024). Large language models and causal inference in collaboration: A comprehensive survey. arXiv preprint arXiv:2403.09606.
>
> •	It is unclear what is "appraisal knowledge xi" in section 2.3.1 or "appraisal context xi" in Algorithm 1.
>
> **Answer**: xi represents appraisal knowledge, which is the same as the appraisal context. We have revised 'appraisal context' to 'appraisal knowledge' to avoid confusion.
>
> •	From Table 2, it is evident that the "AppraisalKnowledge Prompt" is the main contributor to the model performance. I feel like it might be due to the fact that that prompt contains the true emotion label, which in turn makes the appraisal knowledge contain that emotion label (as is the case in Figure 1. The appraisal contains "frustrated" and "angry" which are the emotion labels).
>
> **Answer**: Table 2 provides the ablation study for evaluating only the first component of our framework. It demonstrates that augmenting appraisal knowledge assists LLMs in improving their emotion reasoning during the fine-tuning phase. As noted in Table 2 (line 405), the results are evaluated on the test set. The fine-tuned M_A operates independently, following the principles of cognitive appraisal theory to generate appraisals that lead to inferred emotions. Therefore, no generated appraisal knowledge, including prompts with emotion labels, is provided at this phase. For verification, please refer to our inference-instruction prompt in Appendix B (lines 877-893).

---

> ### Author Response · Authors · 2024-11-20
> **Response to Reviewer 3VTN**
>
> Continue to the Questions:
>
> •	It would be interesting to talk about the multi party nature of most of the real world conversations. And how can this multi-party phenomenon affects the proposed method.
>
> **Answer**: Our third-person appraisal framework is designed to be applicable to multi-party scenarios, as it employs a cognitive-based emotional evaluation that assesses each participant in the conversation from a third-person perspective.
>
> •	Speaker information in the instruction part would also be an interesting thing to see.
>
> **Answer**:  Details such as speakers’ ages or the relationships between speakers are not provided in the available ERC datasets. Future work will consider new data sources.
>
> •	Are the LLMs mentioned in Table 1 also finetuned?
>
> **Answer**: From Table 1, the LLMs corresponding to [5-6] labeled as 'ours' are also fine-tuned.

---

### Official Review · Reviewer_3d1V · 2024-11-02

**Soundness:** 2
**Presentation:** 2
**Contribution:** 2
**Rating:** 5
**Confidence:** 3

**Summary:**

This paper proposes a method to build an emotion evaluation and reasoning dataset for conversations using reflection, along with an offline fine-tuning strategy for emotion evaluation agents based on reinforcement learning. The work introduces the reflexion method from LLMs, where one LLM evaluates emotions while another LLM reflects from a counterfactual perspective - thinking about "Is my judgment correct? Why? How should I judge if the context changes?" and guides the first LLM to update its evaluation according to the reasoning. This approach enables automatic collection of reliable emotion evaluation data in conversational contexts without human annotation. Using this automatically labeled data, the paper presents an offline RL-based fine-tuning strategy. Experiments on IEMOCAP and DailyDialog datasets show improved accuracy and F1 scores in emotion evaluation.

**Strengths:**

1. The proposed automatic data labeling method and offline RL fine-tuning approach show better results in understanding speakers' motivations, attitudes, and goals. This demonstrates the effectiveness of the proposed method.
2. The paper is the first to introduces reflection mechanisms into emotion evaluation, emphasizing self-reflection during the evaluation process. Ablation studies prove the importance of this design choice.
3. As an application-oriented work, the paper demonstrates well-designed experiments, promising results, and clear engineering value.

**Weaknesses:**

1. The paper over-packages its innovations. The core work is simply: a)	applying reflection mechanism to emotion evaluation, b) using reflection to automatically collect high-quality data without human annotation and c) fine-tuning a new evaluation LLM using offline RL with this data. The "Appraisal Generator LLM" and "Appraisal Evaluator LLM" are just LLMs used for reflection, which are essentially the same thing doing self-reflection. The "THIRD-PERSON APPRAISAL AGENT" is just the fine-tuned evaluation LLM. Despite Figure 2's complex appearance, there's no real interaction, feedback loop, or collaborative decision-making between modules. Figure 2 is just a one-way pipeline: M_X—>(M_G<—>M_E)—>M_A.

Here are the suggested revisions for better presenting the paper's contributions:

1) It is suggested to reorganize the Introduction section to clearly state the three core contributions: a) Application of reflection mechanisms to emotion evaluation. b) An automatic data collection method using reflection without human annotation. c) Implementation of an offline RL-based fine-tuning approach with the collected data. Remove excessive theoretical packaging in the Introduction section and focus on engineering improvements and concrete methodological contributions.

2) Simplify the pipeline of the whole approach (Figure 2). The actual data flow is a straightforward pipeline of M_X—>(M_G<—>M_E)—>M_A. It is suggested to remove unnecessary complexity and show the actual data flow more clearly. Try to highlight the practical workflow in each module and between modules.

2. The innovation is limited. While the paper claims to use cognitive science theories and shows fancy formulas and pseudo-code, there's no novel system design, and it uses classic RL methods. The interesting findings are mainly in data collection (using reflection for automatic annotation) and engineering optimization. At its core, it's a solid engineering paper with good practical improvements. But "new method" and "good implementation" are different things. The paper should describe its contributions more honestly and consider submitting to conferences or journals focusing on applied innovations.

Here are the suggested revisions for better presenting the paper's contributions:

1) I suggest you elaborate on the unique value of reflection mechanisms in emotion evaluation. For example, combining specific examples where emotional evaluation is not satisfactory, try to discuss why emotional evaluation performs badly (because of the lack of reflection). Then, emphasize the necessity of introducing reflection in emotional evaluation and reasoning.

2) The "PHASE 1: SELF-EVALUATION" actually is the method for automatically collecting data. It is suggested to reorganize the Sections 2.2 and 2.3.1. If focusing on automatic data collection using reflection, please provide detailed analysis of quality assurance mechanisms in automatic data collection and provide statistic comparison of the cost or quality of collected data with human annotations. The focus should be on the consistency and reliability

3) Based on the offline-RL method in this paper, try to emphasize the difficulty of finetuning the LLM by directly SFT, such as not suitable to handle complex emotional transitions, or anything else. You should explan the reason for choosing RL / offline-RL instead of general training strategy.

4) This work chooses to fine-tune one LLM rather than continue with two frozen LLMS for reflection, obviously hoping for a shorter time consumption. This is a very specific and meaningful choice in the actual scenario. I suggest you highlight your advantage in reasoning efficiency, such as great time savings and no loss of accuracy. Different training methods (such as SFT) have different effects on the accuracy.

3. The paper uses too many mathematical symbols to express simple logical relationships. For example, the basic "try-feedback-improve" process is written in an unnecessarily complex way. While trying to appear more "academic", this actually makes the core ideas harder to understand. Consider using expressions that balance academic rigor with clarity.

Specifically, Algorithm 1 and equiations in Section 2.3.1 should be simplified. I suggest you to draw an figure to represent this feedback loop process. The data collection section mainly provides "how to collect high-quality data", there is no need to worry about how the figure will shorten the length of the content. The novel insight that can be provided is in the data collection process.

**Questions:**

1. Inaccurate or Inconsistent Statements.

    1). For accuracy, suggest changing "To the best of our knowledge, we are the first to integrate counterfactual thinking into a verbal RL-based strategy"  in  line 77-78 to "To the best of our knowledge, this is one of the first works to integrate counterfactual thinking into a verbal RL-based strategy for emotion reasoning tasks". This avoids implying no RL methods have used counterfactual reasoning before, while highlighting the innovation in emotion reasoning and verbal RL.

    2). Regarding contributions, there's an inconsistency between two "first" claims: line 111-112 “To the best of our knowledge, this is the first attempt to apply cognition-based methods to enhance the emotion reasoning capabilities of LLMs in the context of ERC.” and line 77-78 “To the best of our knowledge, we are the first to integrate counterfactual thinking into a verbal RL-based strategy.” Are you referring to the same or different things in these statements?

    3). Sections 2.2 and 2.3.1 are overcomplicated. The simple self-reflection concepts are presented in an unnecessarily complex way.

    4). Line 113-115 “We design an appraisal knowledge prompt, grounded in the principles of cognitive appraisal theory, to generate appraisal knowledge. This approach enables the agent to teach itself how  to reason, thereby addressing existing challenges in ERC” discusses the appraisal knowledge prompt. This shouldn't be listed as a contribution in the Introduction section. It's better presented as part of implementing other innovations. Moreover, since the prompt details are only mentioned in the appendix, it's unsuitable to treat it as an innovation.

2. Too brief and overly general related work.

    1). The section didn’t clarify why adding reflection to emotion reasoning tasks is necessary.  When mentioning research "remains limited” in line 139, what specific limitations exist?

    2). I suggest you to reorganize the content of related work. Specific areas of related work that should be expanded upon and emphase three topics: a) self-reflection/feedback in Emotion Evaluation: previous non-LLM or LLM emotion evaluation method from the pespective of introducing reflection. b) self-reflection in other aspects (such as LLM agent). c) [optional] cognition in LLM agent. If you still want to associate with cognitive theory, you should introduce what work on LLM agent has been further improved by applying reflective / feedback-related cognitive science theories.

    No matter two or three topics, you should explain why directly tranfer the self-reflection in other aspects into emotion evaluation is not enough. What the special issue in the filed of emotion evaluation when using the self-reflection in LLM agents?

3. Writing Clarity Suggestions

    1). To better demonstrate the effectiveness of your proposed methods, it is suggested to reorganize Table 1 into three parts comparing: Mistral (original vs. fine-tuned), Gemma (original vs. fine-tuned) and LLAMA (original vs. causal prompt vs. fine-tuned).

    2). A small mistake: "cause" should be "causal"

    3). The causal prompt details are missing in both the paper and the appendix.  It is suggested to add this setting to the appendix.

4. [Optional] Suggestions to Enhance Innovation

    1). If you plan to release the 600 training examples, this could count as an innovation in data collection. However, the paper doesn't mention building a dataset or any plans to make it public.

    2). What does the dataset collected during the “self-evaluation phase” look like? The paper jumps directly to model inference results from the “meta-evaluation phase” without showing the dataset characteristics.

---

> ### Author Response · Authors · 2024-11-23
> **Response to Reviewer 3d1V**
>
> We sincerely appreciate the reviewer’s thorough and thoughtful review of our paper. Based on the provided suggestions, we have carefully revised the manuscript to address all comments and improve the overall quality. We kindly ask the reviewer to review the updated version and provide any additional feedback. Thank you for your valuable insights and guidance. The following are our responses to the reviewers' comments.
>
> Weaknesses:
> 1.	The paper over-packages its innovations. The core work is simply: a) applying reflection mechanism to emotion evaluation, b) using reflection to automatically collect high-quality data without human annotation and c) fine-tuning a new evaluation LLM using offline RL with this data. The "Appraisal Generator LLM" and "Appraisal Evaluator LLM" are just LLMs used for reflection, which are essentially the same thing doing self-reflection. The "THIRD-PERSON APPRAISAL AGENT" is just the fine-tuned evaluation LLM. Despite Figure 2's complex appearance, there's no real interaction, feedback loop, or collaborative decision-making between modules. Figure 2 is just a one-way pipeline: M_X—>(M_G<—>M_E)—>M_A.
> Here are the suggested revisions for better presenting the paper's contributions:
> 1.	It is suggested to reorganize the Introduction section to clearly state the three core contributions: a) Application of reflection mechanisms to emotion evaluation. b) An automatic data collection method using reflection without human annotation. c) Implementation of an offline RL-based fine-tuning approach with the collected data. Remove excessive theoretical packaging in the Introduction section and focus on engineering improvements and concrete methodological contributions.
>
> **answer**: Thank you for your thorough review and insightful suggestions. We acknowledge that the Introduction section did not effectively communicate our objectives and that our initial presentation of contributions was inadequately framed. To address this, we have reorganized the Introduction to clearly articulate our three main contributions:
>
> 1.	We propose a novel framework that integrates cognitive theory into emotion reasoning tasks, enabling LLMs to autonomously refine their reasoning processes in alignment with cognitive appraisal principles. This is the first work to enhance LLMs’ emotion reasoning capabilities in ERC by guiding them to evaluate emotions based on human cognitive reasoning.
>
> 2.	We incorporate a reflection mechanism to enhance the model’s emotion evaluation in two complementary ways. First, it utilizes counterfactual thinking to generate reflections. Second, it employs the actor-critic RL strategy to improve the model's reasoning capabilities by leveraging these reflections, which serve as a limited number of demonstration examples.
>
> 3.	Experimental results demonstrate that our model enhances prediction performance and generalizability across new dialogue datasets. Additionally, we design an objective method for evaluating emotion reasoning performance, focusing on emotional comprehension, contextual understanding, and expressive coherence and clarity. This evaluation provides a reproducible, explainable, and efficient alternative to manual annotations.
>
> **Please review the updated version provided on lines 112–125.**
>
> 2.	Simplify the pipeline of the whole approach (Figure 2). The actual data flow is a straightforward pipeline of M_X—>(M_G<—>M_E)—>M_A. It is suggested to remove unnecessary complexity and show the actual data flow more clearly. Try to highlight the practical workflow in each module and between modules.
>
> **answer: We have modified Figure 2. Please review the updated version in lines 162–174.**

---

> ### Author Response · Authors · 2024-11-23
> **Response to Reviewer 3d1V**
>
> Continue to the Questions:
> 3.	The innovation is limited. While the paper claims to use cognitive science theories and shows fancy formulas and pseudo-code, there's no novel system design, and it uses classic RL methods. The interesting findings are mainly in data collection (using reflection for automatic annotation) and engineering optimization. At its core, it's a solid engineering paper with good practical improvements. But "new method" and "good implementation" are different things. The paper should describe its contributions more honestly and consider submitting to conferences or journals focusing on applied innovations.
>
> Our paper presents a method that integrates Cognitive Appraisal Theory into a learning framework specifically designed for emotion reasoning in ERC tasks. The **novelty** of our learning framework lies in its ability to enable LLMs to align their reasoning and outputs with the principles of Cognitive Appraisal Theory (CAT), enhancing their capacity for human-like emotion evaluation. This is achieved through a three-component process:
>
> **Appraisal-Knowledge Generation**: Establishes foundational knowledge by linking emotion labels to cognitive appraisal factors, grounding the model's reasoning in CAT principles for subsequent steps.
>
> **Self-Evaluation**: The model generates appraisals according to CAT principles and adjusts its emotional reasoning by reflecting on incorrect examples. This reflective process creates a specialized training dataset that refines the model's understanding and application of CAT principles.
>
> **Meta-Evaluation Process**: Utilizes reinforcement learning to iteratively enhance the model’s reflective capabilities, ensuring continuous refinement of reasoning strategies based on both correct and incorrect examples from the self-evaluation phase. A reflection mechanism is embedded within the RL approach, enabling the model to learn from reflection, thus enhancing the model’s ability of reasoning.
>
> The innovation lies in how these components are synergistically combined to create a unique and effective system for enhancing an LLM’s emotional reasoning capabilities by fine-tuning it on a self-generated dataset. **We have, for the first time, made LLMs capable of emotional reasoning through a self-alignment. This method removes the need for human input and enables the LLM to mimic human-like cognitive appraisal when interpreting emotional states.**
>
> I suggest you elaborate on the unique value of reflection mechanisms in emotion evaluation. For example, combining specific examples where emotional evaluation is not satisfactory, try to discuss why emotional evaluation performs badly (because of the lack of reflection). Then, emphasize the necessity of introducing reflection in emotional evaluation and reasoning.
>
> **answer**:Thank you for your suggestion to elaborate on the unique value of reflection mechanisms in emotion evaluation. To address this, our approach incorporates reflection mechanisms, which provide the following key benefits:
>
> **1.	Learning Reasoning Methods from Both Correct and Incorrect Samples**: By exposing the model to both accurate and flawed reasoning paths, it comprehensively learns effective reasoning strategies. This dual exposure is crucial for handling complex emotional reasoning tasks, enabling the model to recognize and avoid similar mistakes in the future.
>
> **2.	Accelerating Learning with Reflection Mechanisms**: Incorporating reflection allows the model to swiftly identify and correct reasoning errors. This accelerates the learning process, enabling the model to achieve effective reasoning with fewer training samples and enhancing overall learning efficiency.
>
> The "PHASE 1: SELF-EVALUATION" actually is the method for automatically collecting data. It is suggested to reorganize the Sections 2.2 and 2.3.1. If focusing on automatic data collection using reflection, please provide detailed analysis of quality assurance mechanisms in automatic data collection and provide statistic comparison of the cost or quality of collected data with human annotations. The focus should be on the consistency and reliability
>
> **answer:  We have reorganized Sections 2.2 and 2.3.1 by combining them and introducing "Initial Phase" and "Self-Evaluation" to distinguish the content. Please review the updated version on lines 208–229.**
> Although our results were achieved without human annotations, they demonstrate significant improvements, confirming the reliability and effectiveness of our automatic data collection method. Recognizing that human annotation requires time and resources, we plan to incorporate it in future work to further optimize the model’s performance.

---

> ### Author Response · Authors · 2024-11-23
> **Response to Reviewer 3d1V**
>
> Continue to the Questions:
>
> 3.	Based on the offline-RL method in this paper, try to emphasize the difficulty of finetuning the LLM by directly SFT, such as not suitable to handle complex emotional transitions, or anything else. You should explain the reason for choosing RL / offline-RL instead of general training strategy.
>
> **answer:** We chose to use offline RL instead of SFT primarily because RL enables the model to develop reflective capabilities that SFT cannot achieve. Specifically, SFT fine-tunes the model only on correct samples (i.e., accurate emotion appraisals), making it incapable of learning from or understanding the reasons behind errors in its own generated outputs. Consequently, SFT struggles to handle complex emotional transitions effectively.
>
> In contrast, RL allows the model to learn comprehensive reasoning methods by training on both correct and incorrect reasoning paths. By incorporating a reflection mechanism, the model can quickly identify and correct errors, thereby enhancing the accuracy and efficiency of emotional reasoning.
>
> To validate this approach, we conducted experiments on the IEMOCAP dataset, comparing SFT and RL methods. The experimental results are shown in the table below:
>
> Methods.  |  Accuracy.   | Weighted-F1
>
> SFT  |        44.29     |    46.69
>
> RL   |        50.96      |   51.33
>
> This work chooses to fine-tune one LLM rather than continue with two frozen LLMS for reflection, obviously hoping for a shorter time consumption. This is a very specific and meaningful choice in the actual scenario. I suggest you highlight your advantage in reasoning efficiency, such as great time savings and no loss of accuracy. Different training methods (such as SFT) have different effects on the accuracy.
>
> **answer:** Thank you for your suggestion, we have clarified this advantage of using RL instead of SFT.
>
> The paper uses too many mathematical symbols to express simple logical relationships. For example, the basic "try-feedback-improve" process is written in an unnecessarily complex way. While trying to appear more "academic", this actually makes the core ideas harder to understand. Consider using expressions that balance academic rigor with clarity.
>
> **answer:** Our work focuses on academic contribution by presenting a rigorously designed framework that integrates Cognitive Appraisal Theory into LLM-based emotion reasoning. The detailed methodology, including the appraisal-knowledge generation, self-evaluation, and meta-evaluation processes, reflects our commitment to innovation and theoretical rigor. While mathematical formalism is necessary to convey these contributions precisely, we recognize the value of accessibility and will consider adding clarifications to balance clarity with academic depth.
>
> Specifically, Algorithm 1 and equations in Section 2.3.1 should be simplified. I suggest you to draw an figure to represent this feedback loop process. The data collection section mainly provides "how to collect high-quality data", there is no need to worry about how the figure will shorten the length of the content. The novel insight that can be provided is in the data collection process.
>
> **answer: We have simplified Algorithm 1 and the equations in the corresponding section. Please review the updated version in lines 176–190 and 232–258.**
>
> 1). For accuracy, suggest changing "To the best of our knowledge, we are the first to integrate counterfactual thinking into a verbal RL-based strategy" in line 77-78 to "To the best of our knowledge, this is one of the first works to integrate counterfactual thinking into a verbal RL-based strategy for emotion reasoning tasks". This avoids implying no RL methods have used counterfactual reasoning before, while highlighting the innovation in emotion reasoning and verbal RL.
>
> **answer:**  The sentence has been revised.
>
> 2). Regarding contributions, there's an inconsistency between two "first" claims: line 111-112 “To the best of our knowledge, this is the first attempt to apply cognition-based methods to enhance the emotion reasoning capabilities of LLMs in the context of ERC.” and line 77-78 “To the best of our knowledge, we are the first to integrate counterfactual thinking into a verbal RL-based strategy.” Are you referring to the same or different things in these statements?
>
> **answer:** We would like to clarify that these two statements refer to distinct contributions of our work. The first statement (lines 111–112) highlights our novel application of cognition-based methods, specifically Cognitive Appraisal Theory, to enhance the emotion reasoning capabilities of LLMs in the context of ERC. The second statement (lines 77–78) refers to our unique integration of counterfactual thinking into a reinforcement learning-based verbal strategy. These are complementary but separate innovations, each addressing different aspects of our framework.

---

> ### Author Response · Authors · 2024-11-23
> **Response to Reviewer 3d1V**
>
> continue to questions:
>
> 3). Sections 2.2 and 2.3.1 are overcomplicated. The simple self-reflection concepts are presented in an unnecessarily complex way.
>
> **answer:** We acknowledge that Sections 2.2 and 2.3.1 may appear overly complex. To address this, we plan to merge these sections into a single, streamlined section. The revised version will simplify the algorithm, presenting it with clearer and more concise explanations while preserving the essential ideas and maintaining academic rigor. **Please review the updated version in lines 209-228**
>
> 4). Line 113-115 “We design an appraisal knowledge prompt, grounded in the principles of cognitive appraisal theory, to generate appraisal knowledge. This approach enables the agent to teach itself how to reason, thereby addressing existing challenges in ERC” discusses the appraisal knowledge prompt. This shouldn't be listed as a contribution in the Introduction section. It's better presented as part of implementing other innovations. Moreover, since the prompt details are only mentioned in the appendix, it's unsuitable to treat it as an innovation.
>
> **answer:**  We agree with this feedback and have addressed it by removing this part entirely in the revised version.
>
> The section didn’t clarify why adding reflection to emotion reasoning tasks is necessary. When mentioning research "remains limited” in line 139, what specific limitations exist?
>
> **answer:**  We would like to highlight specific limitations related to the absence of self-reflection and feedback mechanisms in current emotion evaluation research.
>
> 2). I suggest you to reorganize the content of related work. Specific areas of related work that should be expanded upon and emphase three topics: a) self-reflection/feedback in Emotion Evaluation: previous non-LLM or LLM emotion evaluation method from the pespective of introducing reflection. b) self-reflection in other aspects (such as LLM agent). c) [optional] cognition in LLM agent. If you still want to associate with cognitive theory, you should introduce what work on LLM agent has been further improved by applying reflective / feedback-related cognitive science theories.
>
> **answer: The Related Work section has been reorganized. Please review the updated version in lines 128–143**.
>
> No matter two or three topics, you should explain why directly transfer the self-reflection in other aspects into emotion evaluation is not enough. What the special issue in the field of emotion evaluation when using the self-reflection in LLM agents?
>
> **answer:** Directly transferring general self-reflection methods to emotion evaluation tasks is insufficient due to the unique challenges in this field. Unlike other tasks, emotional reasoning requires models to identify correct relationships between emotional states and cognitive factors. Simple error reflection often leads to limited improvement because it lacks specific guidance for adjustments.
> To address this, we incorporated counterfactual reasoning into our self-reflection mechanism, enabling the model to utilize its own incorrect predictions as part of the learning process. This approach allows the model to evaluate its errors and explore alternative outcomes that better align with Cognitive Appraisal Theory. As demonstrated in Section 3.3, experiments with standard self-reflection methods showed inferior results compared to our approach. This highlights the effectiveness of our specialized design in addressing the limitations of generic self-reflection for emotional reasoning tasks. Our method leverages counterfactual thinking to bridge the gap, resulting in significant improvements in performance.
>
> To better demonstrate the effectiveness of your proposed methods, it is suggested to reorganize Table 1 into three parts comparing: Mistral (original vs. fine-tuned), Gemma (original vs. fine-tuned) and LLAMA (original vs. causal prompt vs. fine-tuned).
>
> **answer:  The Table 1 has been modified. Please check the updated version in lines 371-377.**
>
> A small mistake: "cause" should be "causal"
>
> **answer:**  This typo has been fixed.
>
> The causal prompt details are missing in both the paper and the appendix. It is suggested to add this setting to the appendix.
> **answer:**  The details of the causal prompt have been added to the appendix for clarity and completeness.
>
> [Optional] Suggestions to Enhance Innovation
> 1). If you plan to release the 600 training examples, this could count as an innovation in data collection. However, the paper doesn't mention building a dataset or any plans to make it public.
>
> **answer:**  Thank you for bringing up this point. We do plan to release training examples upon publication after our paper has been accepted.

---

> ### Author Response · Authors · 2024-11-23
> **Response to Reviewer 3d1V**
>
> continue to questions:
>
> What does the dataset collected during the “self-evaluation phase” look like? The paper jumps directly to model inference results from the “meta-evaluation phase” without showing the dataset characteristics.
>
> **answer:** An example demonstrating how the generated appraisals look and how the LLM adjusts its evaluation during the self-evaluation phase is already included in Appendix B (lines 758–870) of the updated version. According to the dataset generated during self-evaluation, the model employs counterfactual reasoning by generating a hypothetical emotion and assessing whether it aligns with or contradicts the speaker’s goals or expectations. This process helps the model refine its prior understanding. For example: “If the speaker were feeling frustrated, it would suggest that she’s not just seeking a resolution but is also experiencing a sense of exasperation or annoyance with the situation…”

---

### Official Review · Reviewer_bxqR · 2024-11-04

**Soundness:** 3
**Presentation:** 3
**Contribution:** 2
**Rating:** 6
**Confidence:** 4

**Summary:**

This paper introduces an approach to improve emotion recognition capabilities in large language models based on cognitive appraisal theory. The authors develop a well crafted appraisal generation prompt, enabling the model to generate plausible appraisals. Through self-evaluation and meta-evaluation phases, the model is refined to accurately reason about emotions contextually within conversations.
The proposed method results in a model good at generating accurate third-person appraisals, ultimately leading to improved emotion recognition performance.

**Strengths:**

The paper presents a solution for a very relevant problem in dialogue. Whilst LLMs have proven to be very good at many dialogue tasks, they often fail to capture the intricacies in emotion recognition.

The proposed methodology not only improves the emotion recognition ability of the LLM, but ultimately improves the interpretability of the prediction process, which could be very useful when using such emotion recognition models in downstream applications.

**Weaknesses:**

The paper presents a method, which could potentially be interesting to a wider audience and applicable to a wider range of reasoning problems. However, the approach is only tested on a single problem, emotion recognition within dialogue. The paper would be more impactful if it addressed at least one other reasoning task.

The RL approaches in the paper are not novel approaches, but rather the design of the methodology and the prompts designed for appraisal. Hence, it would be beneficial to test this methodology on a wider range of tasks.

**Questions:**

How could this approach be generalised to other tasks? Currently, the prompt is designed for this specific task, and it is not clear how one could easily generate a prompt for other tasks (without prompt engineering and testing).

---

> ### Author Response · Authors · 2024-11-20
> **Response to reviewer bxqR**
>
> Weaknesses:
> The paper presents a method, which could potentially be interesting to a wider audience and applicable to a wider range of reasoning problems. However, the approach is only tested on a single problem, emotion recognition within dialogue. The paper would be more impactful if it addressed at least one other reasoning task.
> The RL approaches in the paper are not novel approaches, but rather the design of the methodology and the prompts designed for appraisal. Hence, it would be beneficial to test this methodology on a wider range of tasks.
>
> Thank you for your valuable feedback.
> Current Emotion Recognition in Conversation (ERC) datasets typically provide emotion-labeled utterances but lack the detailed annotations necessary for in-depth emotion reasoning. To address this, we propose a novel learning framework that enables large language models (LLMs) to autonomously generate training data and refine their reasoning strategies. **This framework is valuable not only for ERC but also for other domains with similar data resource limitations, such as medical diagnosis, legal reasoning, and policy modeling.**
>
> While LLMs show promise in generating domain-specific knowledge, they often face challenges in applying it to complex reasoning tasks. **Our approach integrates cognitive appraisal theory, positioning the LLM as a third-person appraiser to enhance its emotional reasoning capabilities.**
>
> We incorporated **a reflection mechanism into our RL approach**, enabling the model to evaluate both positive and negative outputs generated during the self-evaluation phase and effectively learn how to reflect. The integration of this reflection mechanism into our learning framework, combined with custom-designed prompts tailored specifically for emotion evaluation, represents a unique and valuable contribution to the ERC field.
>
> Our approach provides a versatile solution that can be extended to various reasoning tasks requiring contextual understanding and human-like emotional comprehension.
> We recognize the importance of testing our methodology on a broader range of tasks. In future work, we plan to adapt and evaluate this framework on other emotional reasoning challenges beyond ERC problems.
>
> Questions: How could this approach be generalised to other tasks? Currently, the prompt is designed for this specific task, and it is not clear how one could easily generate a prompt for other tasks (without prompt engineering and testing).
>
> Our prompt design, which incorporates elements such as counterfactual thinking, can be generalized to other tasks by enabling LLMs to engage in reflection without extensive prompt engineering. Moreover, the learning framework we propose is inherently flexible and adaptable to other reasoning tasks, as it allows LLMs to self-align by autonomously generating training data and refining their reasoning strategies without human annotations.

---

### Note · Authors · 2025-02-13

I have read and agree with the venue's withdrawal policy on behalf of myself and my co-authors.

---

### Meta-Review · Area_Chair_bYmb · 2024-12-27

**Metareview:**

Paper is an interesting contribution on improving LLM emotion recognition capabilities via cognitive appraisal theory, but has inaccurate or inconsistent statements, for instance, how the prompt is constructed is different than later claims.

**Additional Comments On Reviewer Discussion:**

NA

---

### Decision · Program_Chairs · 2025-01-22

Reject